# Cyclin-Dependent Kinase Inhibitors in Hematological Malignancies—Current Understanding, (Pre-)Clinical Application and Promising Approaches

**DOI:** 10.3390/cancers13102497

**Published:** 2021-05-20

**Authors:** Anna Richter, Nina Schoenwaelder, Sina Sender, Christian Junghanss, Claudia Maletzki

**Affiliations:** Department of Medicine, Clinic III—Hematology, Oncology, Palliative Medicine, Rostock University Medical Center, 18057 Rostock, Germany; Anna.Richter@med.uni-rostock.de (A.R.); Nina.Schoenwaelder@med.uni-rostock.de (N.S.); Sina.Sender@med.uni-rostock.de (S.S.); Christian.Junghanss@med.uni-rostock.de (C.J.)

**Keywords:** CDK4/6 inhibitors, predictive biomarker, mechanisms of resistance, pharmacological inhibition, combination strategies

## Abstract

**Simple Summary:**

Cyclin-dependent kinases are involved in the regulation of cancer-initiating processes like cell cycle progression, transcription, and DNA repair. In hematological neoplasms, these enzymes are often overexpressed, resulting in increased cell proliferation and cancer progression. Early (pre-)clinical data using cyclin-dependent kinase inhibitors are promising but identifying the right drug for each subgroup and patient is challenging. Certain chromosomal abnormalities and signaling molecule activities are considered as potential biomarkers. We therefore summarized relevant studies investigating cyclin-dependent kinase inhibitors in hematological malignancies and further discuss molecular mechanisms of resistance and other open questions.

**Abstract:**

Genetically altered stem or progenitor cells feature gross chromosomal abnormalities, inducing modified ability of self-renewal and abnormal hematopoiesis. Cyclin-dependent kinases (CDK) regulate cell cycle progression, transcription, DNA repair and are aberrantly expressed in hematopoietic malignancies. Incorporation of CDK inhibitors (CDKIs) into the existing therapeutic regimens therefore constitutes a promising strategy. However, the complex molecular heterogeneity and different clinical presentation is challenging for selecting the right target and defining the ideal combination to mediate long-term disease control. Preclinical and early clinical data suggest that specific CDKIs have activity in selected patients, dependent on the existing rearrangements and mutations, potentially acting as biomarkers. Indeed, CDK6, expressed in hematopoietic cells, is a direct target of MLL fusion proteins often observed in acute leukemia and thus contributes to leukemogenesis. The high frequency of aberrancies in the retinoblastoma pathway additionally warrants application of CDKIs in hematopoietic neoplasms. In this review, we describe the preclinical and clinical advances recently made in the use of CDKIs. These include the FDA-approved CDK4/6 inhibitors, traditional and novel pan-CDKIs, as well as dual kinase inhibitors. We additionally provide an overview on molecular mechanisms of response vs. resistance and discuss open questions.

## 1. Introduction

Despite the clinical implementation of novel treatments like kinase inhibitors, hypomethylating agents or pathway modulators during the last decades, several hematological malignancies still face a poor prognosis. Relapsed or refractory courses are particularly difficult to manage, often exhibiting multiple drug resistances and intimidating survival rates [1,2,3,4]. The FDA approval of the cyclin dependent kinase (CDK) 4/6 inhibitors (CDKIs) palbociclib, ribociclib, and abemaciclib for hormone receptor positive and human epidermal growth factor receptor 2 negative locally advanced or metastatic breast cancer treatment led to numerous promising clinical studies investigating CDKIs in solid neoplasms. This provides a rationale for the emerging intensive testing in hematological malignancies, raising hopes for improving patients’ prognoses. Given their wide spectrum of cellular modulation including cell cycle control, transcription, DNA repair, epigenetic regulation, proliferation, and apoptosis, CDKs represent promising molecular targets for leukemia and lymphoma treatment [5].

Cell cycle regulation is a key mechanism to prevent malignant cell proliferation and uncontrolled cell division. Aleem and Arceci reviewed the roles of CDKs in controlling cell cycle and development of hematological malignancies in detail [6]. Here, we focus on present and future therapeutic approaches to overcome CDK-influenced cell cycle control malfunction (Appendix A) and will thus only offer a short introduction on CDK pathways and their role on leukemogenesis. In brief, cell cycle control is mainly mediated by serine/threonine CDKs acting as catalytic subunit when activated by their respective cyclins. CDK activity is further modulated by physiological CDKIs and posttranslational modification, resulting in transcriptional regulation, DNA damage repair mechanisms, metabolism, or epigenetic processes. In addition to the “classical” CDKs directly influencing the cell cycle, further kinases act as indirect modulators to regulate transcription or epigenetic signaling [6].

Hematopoietic stem cells are relatively quiescent to prevent stem cell exhaustion [7,8,9]. Once these stem cells initiate cell cycle induction, the cells proliferate extensively to provide hematopoiesis [10]. Therefore, the cell cycle of hematopoietic stem cells must be controlled thoroughly. Dysregulation of CDKs and associated cyclins is frequently observed in hematological malignancies. CDK6 is predominantly expressed in hematopoietic cell types and loss of CDK6 results in impaired generation of several blood cell types. In contrast, overexpression and chromosomal translocation of CDK6 is observed in acute lymphoblastic leukemia (ALL) and lymphoma. CDK4/6 inhibition is achieved by physiological CDKI p16*^INK4A^*, the most frequently deleted locus in human cancer. Translocations of the *MLL* gene locus are common in acute myeloid leukemia (AML) and account for most infant ALL cases. CDK6 is a direct target of MLL fusion proteins and thus activated. Finally, CDK6 can be activated by *FLT3*-ITD-mediated down-regulation of cyclins D2 and D3. Besides CDK6, other CDKs are also involved in leukemogenesis. For example, the AML driver mutation *FLT3*-ITD is an activator of CDK1. Further, mutations and deletions of the cyclin C and CDK19 locus on 6q21 result in altered Notch1 regulation especially in T-ALL [6].

## 2. CDK4/6 Inhibitors

While the pan CDKI flavopiridol was the first CDK inhibitor applied clinically [11], recent CDKIs are more specific, with most drugs targeting only a subset of CDKs. CDK6 has a central role in hematopoiesis and CDK6-deficient mice show reduced production of erythrocytes, granulocytes, macrophages, neutrophils, and thrombocytes, as well as thymic atrophy [6]. Neutropenia is therefore the most common and dose-limiting adverse event in the clinical use of CDK6 inhibitors [12]. CDK6 is also a direct target of MLL fusion proteins which are common in AML and ALL [13,14]. This results in transcriptional activation of CDK6 and subsequent initiation of leukemic processes [14,15,16]. Further, CDK6 can be activated via *FLT3*-ITD-mediated upregulation of cyclins D2 and D3 [17]. In mantle cell lymphoma, the characteristic t(11;14) translocation induces ectopic cyclin D1 expression, also resulting in CDK6 and CDK4 upregulation [18]. In pediatric B-ALL, p16*^INK4A^* deletions, especially occurring during relapse, are a key feature of dysfunctional CDK4/6 control and an associated dismal prognosis [19].

### 2.1. Palbociclib

The CDK4/6 inhibitor palbociclib was widely evaluated in solid tumors and is now also analyzed in a variety of hematopoietic malignancies. It demonstrated significant pre-clinical in vitro and in vivo efficacy in AML cells with *FLT3*-ITD [17,20] and TKD mutations [21], *RUNX1/ETO* translocation [22] and *MLL* rearrangement [15,23]. In a recent study investigating de novo transformation of granulocyte/macrophage progenitor cells to AML, Chen et al. demonstrated that transient palbociclib application is capable of halting progenitor cell proliferation and preferentially abrogated the most proliferative progenitor cell subsets. Palbociclib further reduced the progenitor cell transformation in vivo, resulting in reduced AML burden and prolonged survival. This suggests that cell cycle inhibition decreases the likelihood of malignant transformation in vivo [24].

In T-ALL cells as well as B-ALL cells featuring *MLL* or BCR-ABL1 rearrangements, palbociclib controlled cell growth via G1 arrest and Rb dephosphorylation both in vitro and in vivo [14,25,26,27,28]. A phase I clinical trial investigating palbociclib in relapsed ALL children is currently underway.

Similar promising antitumoral effects were observed in multiple myeloma, anaplastic large-cell lymphoma, and mantle cell lymphoma, where palbociclib was also capable of overcoming ibrutinib resistance [29,30,31,32,33,34]. In a study with 17 relapsed mantle cell lymphoma patients, palbociclib achieved one complete remission (CR) and two partial responses (PR), five patients had a progression-free survival of at least one year with reduced tumor metabolism and proliferation [35]. A phase I study in non-Hodgkin lymphoma showed stable disease (SD) in one third of the participants and two out of 68 patients had PR [36].

Palbociclib was then evaluated in several combinations both preclinically and clinically. The combination with proteasome inhibitor bortezomib was effective in an immunocompetent myeloma mouse model. Inhibition of CDK4/6 by palbociclib induced G1 arrest and enhanced bortezomib susceptibility via increased mitochondrial depolarization [30]. This combination was also evaluated in a clinical phase I/II trial in refractory/relapsed (R/R) myeloma together with dexamethasone, demonstrating an overall response rate of 20% and SD in 44% of the participants [37]. A phase I trial in R/R mantle cell lymphoma with palbociclib and bortezomib achieved CR in one out of 19 patients [38]. Combined palbociclib and ibrutinib treatment has been evaluated in R/R mantle cell lymphoma. From 27 patients in this phase I study, 67% responded and 37% had CR [39].

Further combination partners are evaluated preclinically in vitro and in vivo. In AML, the cytarabine dose could be reduced after palbociclib priming (Figure 1) [40]. Besides cell cycle regulation, CDK6 also controls gene expression of oncogenic kinases by directly binding promoter sites. These include, among others, AURORA and AKT, both of which are known mediators of drug resistance. *FLT3* mutations can further contribute to pathway activation in AML. Combined palbociclib and pan-AURORA kinase inhibitor danusertib or AKT inhibitor MK-2206 treatment resulted in synergistic anti-leukemic effects in *FLT3*-ITD and TKD mutated AML cells [21].

Although there is a clear connection between BCR-ABL1 fusion and cell cycle regulation, only very limited data is available for the evaluation of CDKIs in chronic myeloid leukemia (CML). Rangatia and Bonnet have shown that lack of BCR-ABL1 leads to G1 phase arrest and a decrease in cyclin D1. Physiological CDKIs p21 and p27 subsequently exhibited increased gene expression [41]. Schneeweiss-Gleixner et al. recently reported that palbociclib synergizes with tyrosine kinase inhibitor ponatinib in BCR-ABL1*^T315I^* mutated CML, via reducing proliferation and inducing G1 arrest. This is of importance because tyrosine kinase inhibitor resistance is frequently observed in this CML subtype, leading to challenging clinical problems [42]. In ALL, palbociclib synergizes with fibroblast growth factor receptor 1 inhibitor PD-173074 and imatinib (Gleevec) [43,44]. Increased apoptosis rates compared to palbociclib mono application were achieved with PI3Kδ inhibitor GS-1101 and BET protein bromodomain antagonist JQ1 in mantle cell lymphoma [45,46]. In myeloma and diffuse large B-cell lymphoma, synergism was achieved with dexamethasone and Bruton’s tyrosine kinase inhibitor tirabrutinib [29,47].

To demonstrate their anti-leukemic potential, CDK4/6 inhibitors rely on expression of downstream signaling protein Rb. Intrinsic or acquired lack of Rb function results in resistance towards therapeutic CDK4/6 targeting [12]. Additionally, palbociclib treatment can lead to p27 downregulation, thus reactivating CDK2 and cell cycle progression. Transcription factor *FOXO3A* can additionally control *p27* gene expression. However, it was not influenced in palbociclib-resistant cells; ruling out the possibility of *FOXO3A* directly induced *p27* downregulation. Palbociclib-sensitive cell lines exhibited a similar drop in *p27* gene expression, while protein abundance remained stable. Hence, the *p27*-downregulating effect is rather due to posttranslational modification than reduced *p27* gene expression (Figure 2) [17].

### 2.2. Ribociclib

Ribociclib (LEE011) is another CDK4/6 inhibitor that demonstrated significant anti-proliferative and apoptosis-inducing effects in AML and B-ALL cell lines and primary samples, probably mediated via G1 arrest and senescence [48]. A recent study on pediatric B-ALL evaluated ribociclib in combination with dexamethasone. They found significant basal overexpression of CDK6 in B-ALL cells. Subsequent inhibition of CDK4/6 using ribociclib resulted in G1 phase arrest, reduced cell proliferation, and apoptosis induction accompanied by Rb dephosphorylation. Adding dexamethasone synergistically improved anti-neoplastic effects in B-ALL cell lines as well as primary samples [49].

Pikman et al. investigated ribociclib in *NOTCH1*-mutant and wildtype T-ALL and found both subtypes sensitive to the inhibitor. The combination with glucocorticoid dexamethasone and mTOR inhibitor everolimus acted synergistically in vitro and in vivo. In contrast, antagonistic effects were observed with several drugs used in T-ALL standard chemotherapy including methotrexate, mercaptopurine, asparaginase, or doxorubicin [50].

In T-ALL, CDK6 is required for AKT- or Notch1-induced leukemia initiation. Jena et al. demonstrated increased *CD25* and *RUNX1* expression upon CDK6 inhibition and that CD25 ablation results in T-ALL leukemogenesis. They further showed that CD25 mediates the therapeutic response to ribociclib, suggesting CD25 deletion as a potential mechanism of resistance to CDK6 inhibitors as well as predictive marker for ribociclib response (Figure 1) [51]. A clinical phase I study examined ribociclib in solid neoplasms and lymphoma and found SD in ten out of 70 patients [52].

### 2.3. Abemaciclib

While ribociclib was mainly analyzed in acute leukemias, abemaciclib (LY2835219) has been more widely evaluated in lymphoma. Here, inhibited proliferation and induced apoptosis was reported in germinal center-derived B-cell lymphoma cell lines, but not in activated B-cell like diffuse large B-cell lymphoma cell lines [53]. In a clinical trial, 71% of the participating R/R mantle cell lymphoma patients achieved SD and 36% had PR [54]. A subsequent phase II study with 28 patients reached CR in two patients and an overall response rate of 36% with an overall survival of 16 months [55]. Another clinical trial is currently evaluating abemaciclib with fulvestrant in advanced or metastatic solid tumors and lymphomas.

Further preclinical studies found abemaciclib effective in *MLL*-rearranged AML cell lines and xenografts [56] as well as myeloma cell lines [57]. Nakatani et al. recently identified that t(8;21) rearranged AML cell lines are more vulnerable to palbociclib and abemaciclib than non-t(8;21) AML cells [58]. Molecularly, this is due to higher cyclin D2 levels, a common response marker towards abemaciclib. In t(8;21) rearranged AML cells, abemaciclib induced the expected G1 arrest, leading to impaired cell proliferation and decreased MAPK and AKT pathway signaling. Another interesting finding of this study was autophagosome formation which significantly increased apoptosis induction upon combined application of abemaciclib and autophagy inhibitors [58].

### 2.4. Lerociclib

The novel CDK4/6 inhibitor lerociclib (G1T38) was capable of decreasing Rb phosphorylation and induced G1 cell cycle arrest in leukemia and lymphoma cell lines in vitro. Of note, lerociclib demonstrated a superior efficacy compared to palbociclib and did not induce severe neutropenia in an estrogen receptor positive breast cancer dog animal model, which is a common side effect of CDK4/6 inhibitors. Using mouse xenograft models, the authors further demonstrated that lerociclib accumulated within the tumor but not in plasma, implying less severe effects on myeloid progenitor cells compared to palbociclib therapy. Indeed, plasma concentrations after palbociclib treatment were significantly higher than the concentration needed to inhibit cell proliferation in several cell lines. In contrast, after lerociclib treatment, plasma concentrations dropped faster. The authors thus claim that palbociclib-induced neutropenia is a product of CDK4/6 inhibition in the bone marrow, preventing proliferation of healthy bone marrow cells [59].

CDK4/6 inhibitors including the novel promising molecule lerociclib are without doubt the most intensively researched group of CDKIs in both, solid and hematological malignancies. While palbociclib, ribociclib, and abemaciclib already obtained FDA approval for breast cancer, current preclinical and clinical studies for leukemia and lymphoma also focus on inhibitors targeting other CDKs.

## 3. CDK7/8/9 Inhibitors

CDKs 7, 8, and 9 are novel emerging targets in preclinical research. Indeed, recent data reveal anti-leukemic activity upon inhibition of these transcriptional regulators. CDK9 is a global transcriptional regulator and part of the super elongation complex controlling RNA polymerase II phosphorylation and elongation. AFF family members, which are frequently fused to *MLL* in acute leukemias, are also part of the super elongation complex, thus mis-localizing CDK9 to *HOX* gene promoters and inducing abnormal CDK9 expression, cell growth, and proliferation [60]. Another role for CDK9 is transcriptional regulation of MCL-1, it may thus influence intrinsic apoptosis induction (Figure 3) [61].

### 3.1. AZD4573

So far, AZD4573 is the only selective CDK9 inhibitor invested clinically in hematological malignancies. A preclinical study on different hematological neoplasms proved significant induction of apoptosis via MCL-1 suppression in vitro and in vivo [62]. The same study also evaluated the BCL-2 inhibitor venetoclax as combination partner to overcome treatment failure or leukemic progress after AZD4573 cessation. Both, MCL-1 and BCL-2 are anti-apoptotic members of the intrinsic apoptosis initiation cascade. Combined therapy with AZD4573 and venetoclax induced long-term inhibition of leukemic cell proliferation in all animals, even in models intrinsically resistant to either monotherapy (diffuse large B-cell lymphoma SU-DHL-4; AML model OCI-AML3) [62].

### 3.2. CDKI-73

The non-selective CDK9 inhibitor CDKI-73 proved anti-proliferative and pro-apoptotic capacity in chronic lymphoblastic leukemia (CLL) [63], diffuse large B-cell lymphoma [64], ALL and AML [65] cells, as well as in animal models. Walsby et al. evaluated the anti-neoplastic potential of CDKI-73 together with fludarabine in CLL and found decreased *MCL1* gene expression after CDK9 inhibition while fludarabine had the opposite effect. Combined application of CDKI-73 and fludarabine downregulated RNA polymerase II mediated genes *MCL1*, *XIAP*, *CCND1*, and *CCND2*. Of note, synergistic effects were also observed under CD40L-expressing pro-survival co-culture conditions with initial fludarabine resistance [63]. In diffuse large B-cell lymphoma, CDK9 inhibitors including CDKI-73 frequently induce histone 3 lysine 27 trimethylation, which is associated with tumor progression. A recent study therefore evaluated CDKI-73 with histone methyltransferase EZH2 inhibitors EPZ6438 and GSK126 and found synergistic anti-proliferative effects. The authors also identified drastically increased apoptosis and DNA damage in response to combined treatment [64]. The same group previously evaluated combinatorial effects of CDKI-73 with venetoclax in ALL and AML and elucidated synergistic induction of apoptosis via PARP and caspase 3 cleavage as well as XIAP downregulation [65]. Many more CDK9 inhibitors have been preclinically evaluated in a broad range of hematological malignancies [66,67,68,69,70,71].

### 3.3. CDK8 Inhibitors

CDK8 regulates transcription as part of the mediator complex or by phosphorylation of transcription factors [72]. In primary AML and ALL samples with BCR-ABL1 translocation, Menzl et al. found that mTOR signaling is usually deregulated in CDK8-deficient cells. They subsequently developed the small molecule YKL-06-101, targeting both CDK8 and mTOR, which had significant anti-leukemic potential in vitro and in vivo [73]. Another selective CDK8 inhibitor, SEM120, is effective in AML in vitro and in vivo and currently evaluated in a phase Ib clinical trial in AML and myelodysplastic syndrome [74].

### 3.4. CDK7 Inhibitors

First CDK7 inhibitors are also arising and are being tested in AML cell lines. The CDK7/12/13 inhibitor THZ1 showed anti-proliferative effects and induced apoptosis in a *RUNX1/ETO* rearranged cell line [75]. A preclinical study investigating the effect of THZ1 in myeloma revealed decreased cell proliferation and survival as well as RNA polymerase II, CDK 1, 2, and 9, MCL-1, BCL-XL, and c-MYC downregulation in vitro and in vivo. Addition of venetoclax or carfilzomib significantly increased the antitumor efficacy [76].

SY-1365, which is under clinical investigation for solid tumors, decreased MCL-1 protein levels and induced transcriptional changes mainly of oncogenic transcription factor genes and members of cell cycle and DNA damage repair related pathways. Of note, this inhibitor was more effective in cells with low BCL-XL expression. When combined with BCL-2 inhibitor venetoclax, antitumor effects were synergistically increased in vitro and in vivo [77]. Finally, BS-181 inhibited CDK7 and induced apoptosis in Jurkat T-ALL cells [78].

Collectively, CDK7/8/9 inhibitors hold promise for being implemented in trials because of their unique ability to modify transcriptional processes and regulate apoptosis. Other approaches featuring down-regulation of these kinases are less specific, resulting in a broader spectrum of mechanistic actions to be dealt with.

## 4. Pan CDK Inhibitors

### 4.1. Flavopiridol

Flavopiridol (alvocidib) is a first generation pan CDK inhibitor, targeting CDKs 1, 2, 4, 6, 7, and 9. It induces cell cycle arrest in ALL and AML cell lines as well as leukemia and lymphoma animal models [79,80]. Preclinical studies of flavopiridol in combination with BCL-2 inhibitor venetoclax or pan BH3 mimetic obatoclax revealed synergistic anti-apoptotic effects in vitro and in vivo, probably mediated via MCL-1, BIM, and NOXA regulation [81,82]. Flavopiridol has also been tested in CML and acted synergistically with pro-apoptotic pyrrolo-1,5-benzoxazepine compounds in imatinib-resistant cells. The observed induction of apoptosis was likely due to deactivation of the CDK1/cyclin B1 complex [83].

Clinical efficacy was observed in hematological neoplasms including CLL and AML, especially in AML as part of the FLAM regimen (flavopiridol followed by cytarabine and mitoxanthrone) [6,84]. Clinical investigation in CLL with early trials using flavopiridol achieved PR in almost half of the patients [85]. Further, a clinical trial is currently evaluating the potential of flavopiridol and decitabine in myelodysplastic syndrome. However, the use of flavopiridol proved difficult in the clinical setting, offering a complex pharmacokinetic profile, a wide range of side effects and an unclear mechanism of action [12].

Several mechanisms of resistance have been described for flavopiridol (Figure 2). Mahoney et al. identified endoplasmatic reticulum stress-mediated death of CLL cells as a novel mode of action for flavopiridol. However, induction of autophagy decreased cytotoxic effects while autophagy inhibition supported stress-mediated anti-leukemic effects, highlighting autophagy as a potential mechanism of flavopiridol resistance [86]. Besides, members of the BCL-2/MCL-1 apoptosis signaling cascade are involved in flavopiridol resistance. Data by Yeh et al. indicated that prolonged MCL-1 stability, in line with RNA polymerase II phosphorylation and CDK9 kinase domain upregulation, contributes to resistance in B-ALL cell line 697. MCL-1 upregulation was probably mediated via MAPK/ERK signaling. Knockdown of MCL-1 restored flavopiridol sensitivity and induced cytotoxicity [87]. Besides MCL-1, BCL-2 is another anti-apoptotic signaling molecule involved in flavopiridol metabolism. Decker et al. demonstrated that a truncated BCL-2 protein lacking the phosphorylation loop domain results in flavopiridol resistance in U937 AML cells [88].

### 4.2. Dinaciclib

Dinaciclib (SCH727965) is now widely evaluated in hematological malignancies. By inhibiting CDKs 1, 2, 5, and 9, dinaciclib reduces cell viability, induces apoptosis and cell cycle arrest in ALL and AML cell lines, also with *MLL* rearrangement, primary patient cells, and in vivo models [89,90,91,92]. A phase II study found reduced numbers of circulating blasts but no bone marrow remission in AML and ALL patients [91].

In CLL, dinaciclib downregulates MCL-1 gene and protein expression and potently induces apoptosis in patient-derived cells [93]. Interestingly, this effect was also present when the cells were cultured with cytokines produced by microenvironment cells but not with direct stromal cell contact. This lack of therapeutic efficacy was overcome by addition of PI3Kα inhibitor PIK-75 but not inhibitors of other PI3K isoforms [93]. Chen et al. subsequently characterized dinaciclib-induced effects on signaling pathways, elucidating caspase 8 and 9-mediated apoptosis induction with MCL-1 and BCL-XL suppression. They further detected inhibition of oncogenic pathways including STAT3, NFkB, p38, PI3K/AKT, and MAPK [94]. In addition, dinaciclib enhanced ibrutinib and venetoclax sensitivity. When combined with SYK inhibitor entospletinib, no synergistic effect was seen. Two phase I clinical trials evaluated the effect of dinaciclib in R/R CLL; combined with rituximab, four out of five patients had SD and one achieved CR [95]. In combination with ofatumumab, a median progression-free survival of 322 days was observed [96].

In mantle cell lymphoma, Höring et al. pursued a combined MCL-1 inhibiting and NOXA stabilizing approach using dinaciclib and fatty acid synthase inhibitor orlistat. They observed synergistically-induced NOXA-dependent apoptosis in mantle cell lymphoma cell lines and primary samples along with tumor growth inhibition in vivo [97]. In advanced non-Hodgkin lymphoma, B-CLL, and myeloma, no PR, CR, or SD was observed in a clinical phase I study with dinaciclib alone [98]. A different study reported SD in eight out of 61 advanced non-Hodgkin lymphoma or myeloma patients treated with dinaciclib in combination with aprepitant, ondansetron, and dexamethasone [99].

Further preclinical evaluation detected increased doxorubicin response rates in myeloma cell lines after low dose dinaciclib treatment, boosting growth inhibition and promoting senescence [100]. Interestingly, and in contrast to several other studies of dinaciclib in hematological neoplasms, this work did not observe apoptosis induction but accelerated senescence via increased p16 signaling. Additionally, dinaciclib may have reduced doxorubicin-induced ATM/Chk2/p53/p21 senescence-modulating signaling. Further investigation in myeloma was conducted by Alagpulinsa et al. who combined dinaciclib with PARP inhibitor ABT-888, based on the idea of impaired homologous recombination during PARP-mediated DNA double strand break repair after dinaciclib [101]. Dinaciclib-treated myeloma cells had increased DNA damage and reduced repair gene expression. Cotreatment with PARP inhibitor ABT-888 synergistically reduced tumor cell proliferation in vitro as well as in vivo using myeloma xenograft models. A subsequent phase I/II study found an overall response rate of 11% in myeloma patients receiving dinaciclib alone [102]. A combinatory approach with bortezomib is currently being evaluated in a phase I plasma cell myeloma study.

### 4.3. Other Pan CDK Inhibitors Investigated in Clinical Trials

Further pan CDK inhibitors are now also tested in the clinical setting. Besides the CDKs targeted by dinaciclib (CDK1, 2, 5, 9), AT7519M also inhibits CDK4, still inducing apoptosis in vitro and in vivo and reducing proliferation in myeloma and CLL [103,104]. Two clinical trials investigating AT7519M in non-Hodgkin lymphoma as well as R/R CLL and mantle cell lymphoma observed SD in at least half of the patients enrolled and PR in three out of 12 R/R mantle cell lymphoma cases [105,106].

Voruciclib (P1446A), a CDK1/2/4/5/6/8/9 inhibitor, was evaluated in a phase I study recruiting follicular lymphoma, mantle cell lymphoma, marginal zone lymphoma, small lymphocytic lymphoma, CLL, diffuse large B-cell lymphoma, and AML patients after demonstrating significant potential in preclinical studies [61,107]. Combined application of voruciclib and BCL-2 inhibitor venetoclax was synergistic in preclinical AML models. Notably, venetoclax-mediated *MCL-1* and *c-Myc* downregulation was only reached in an intermittent drug administration schedule that should be considered in clinical practice [108].

The CDK1/4/9 inhibitor P276-00 showed promising in vitro and in vivo results in AML [109], myeloma [110], and mantle cell lymphoma [111] but only achieved SD in two out of 13 R/R mantle cell lymphoma patients in a phase II clinical trial [112].

Roniciclib (BAY1000394) targets the same CDKs as flavopiridol (1, 2, 4, 6, 7, and 9) and achieved SD in three out of seven patients with lymphoid neoplasms [113]. Of note, this is the only clinical trial so far investigating CDKIs in classical Hodgkin lymphoma. Earlier studies suggested that there is a relation between apoptosis-induction, DNA fragmentation, survival, and expression of cell cycle regulators p27 and p21 [114], providing a rationale for CDK targeting in Hodgkin lymphoma. Sánchez-Aguilera et al. further identified CDK regulator p18*^INK4C^* as a potential tumor suppressor gene in Hodgkin lymphoma [115]. Finally, loss of p16*^INK4A^* expression has been observed in Reed–Sternberg cells of Hodgkin lymphoma cases [116].

Due to its variety in mechanistic pathways, pan CDK inhibition offers a broad range of possibilities to target virtually every hematological subtype. Most progress has been made for dinaciclib, which is already investigated in clinical trials; however, results are ambiguous. Chemical modifications or combinatorial schedules might be beneficial to increase the therapeutic outcome.

## 5. Dual Kinase Inhibitors and Novel Approaches

Developing resistance to CDKI monotherapy is frequently seen in both preclinical and clinical studies. Hence, novel approaches are being evaluated, including dual kinase inhibitors. In the AML setting, most inhibitors target one or more CDKs and FLT3. AMG925 and FN-1501 inhibit FLT3 in combination with CDK4 and CDK2/4/6, respectively. Both drugs significantly induced apoptosis and anti-leukemic effects as well as ERK/AKT/Rb dephosphorylation in vitro and in vivo [117,118].

TG02, a pan CDK and FLT3 inhibitor, further targets JAK2. This inhibition promoted G1 arrest, apoptosis, and tumor regression in AML cell lines, in vivo models, and primary samples [119,120,121]. TG02 is also effective in myeloma cell lines as single agent under protective bone marrow niche conditions and in xenografts. TG02 further demonstrated synergistic potential with approved anti-myeloma agents dexamethasone, melphalan, bortezomib, and lenalidomide, possibly via ERK5 blockade, intrinsic and extrinsic apoptosis induction, and cell cycle blockade [122].

Additionally in myeloma, pan CDK/JAK/Src/AMPK/GSK3β inhibitor RGB-286638 achieved cytotoxicity in vitro and prolonged animal survival [123]. Besides CDK4 and CDK6, ON123300 and ON108110 inhibit PI3Kδ and CK2, respectively, and induced Rb dephosphorylation, G1 arrest, and apoptosis in mantle cell lymphoma [124,125].

Structural similarity of CDK ATP binding sites is a major challenge in the design and development of selective inhibitors. Proteolysis-targeting chimeras (PROTACs) are a novel approach of CDK elimination. These synthetic molecules comprise a ligand for the protein of interest, for example, CDK6, and another ligand which recruits E3 ligases. E3 ligases then induce ubiquitination and proteasomal degradation of the target structure. In a recent study of De Dominici et al. investigating BCR-ABL1-positive ALL, PROTAC YX-2-107 was designed to bind CDK4/6, reducing CDK6 enzymatic activity and inducing degradation in vitro. Further, S phase transition was suppressed and Rb and FOXM1 dephosphorylation was observed. In vivo, PROTAC YX-2-107 achieved a comparable or better reduced leukemia burden in PDX mice compared to palbociclib [126]. Another PROTAC, ARV-771, targets BET and acts synergistically with palbociclib in ibrutinib-resistant mantle cell lymphoma cells [127]. PROTACs have also been designed to inhibit other CDKs. Qiu et al. designed a series of PROTACs based on CDK9 inhibitor atuveciclib (BAY-1143572) and showed anti-leukemic effects at low nanomolar concentrations, also in vivo [128].

Novel and innovative strategies like dual kinase inhibition or PROTAC design are important steps towards an improved targeted therapy. Further preclinical studies in syngeneic and xenograft models are needed to identify candidates with a high likelihood of proving beneficial in a clinical setting.

## 6. The Interaction of CDKIs with the Tumor Microenvironment

A major challenge for receiving long-term disease free-survival in hematological diseases is successful targeting of the malignant niche in the bone marrow. The latter is a complex microenvironment in which hematopoietic stem cells interact with multiple non-hematopoietic cell types. Just like normal stem cells, leukemia stem cells, hosted in the stem cell niche, undergo self-renewal, can efflux drugs, and have a quiescent cell cycle status, which makes them difficult to target. Especially CDK6 plays a critical role in hematopoietic stem cell differentiation [129]. A comparable role in leukemia stem cells can be expected.

Ischemia-like conditions are the driving force of leukemia stem cell refractoriness to classical drugs as well as targeted agents, such as imatinib mesylate [130]. Though experimental evidence for the interaction of CDKIs with hypoxic niches of the bone marrow is still pending, experiences from solid tumor models described successful reversal of hypoxia-mediated therapy resistance via CDK1 and CDK2 interaction with hypoxia inducible factor-1 [131]. The mechanism was due to Rb hypophosphorylation by palbociclib, which favors cell cycle arrest and senescence. A recent case report identified rebound lymphocytosis in a CLL patient after terminating palbociclib treatment for synchronic breast cancer [132]. This report impressively demonstrates the interaction of CDK4/6 inhibitors with the tumor microenvironment via induction of a cell cycle arrest. Still, follow-up studies should focus on the stem cell niche to judge the efficacy of CDKIs and ultimately prevent relapse.

## 7. Conclusions

The field of hematological neoplasms is very heterogeneous with each entity featuring individual characteristics and challenges. However, most of them are still difficult to treat, indicated by high relapse rates, intimidating prognoses, and lack of durable curative therapies. CDKs and their regulating cyclins are of major importance for the tumorigenic potential of developing leukemia or lymphoma cells and frequently dysregulated in those malignancies. Modulation of CDK expression and activity thus represents a promising strategy to target aberrant cell cycle progression and proliferation in hematological tumor cells.

The results of extensive preclinical evaluations are promising, with several clinical studies currently testing CDKIs in most hematological malignancies. CDK4/6Is palbociclib, abemaciclib, and ribociclib are already FDA-approved for solid neoplasms, raising hopes for a subsequent clinical implementation in hematological malignancies in the near future. Interestingly, clinical trials have mainly been finished for lymphoma (especially mantle cell lymphoma) and CLL patients while data on acute leukemias are lagging behind. Published results indicate that combined therapy with other small molecule inhibitors like ibrutinib or bortezomib as well as inhibitors of anti-apoptotic proteins may have the capacity to circumvent CDKI resistance development. Importantly, most clinical studies so far have been conducted in heavily pretreated relapsed/refractory patient cohorts, highlighting the potential of CDK targeting for hematological malignancy management. 

Another open question that must be addressed in future studies is why inhibitors of the same class result in highly variable response rates in the same entity. This underlines the importance of further investigation on biomarkers, resistance mechanisms, and exact modes of action.

This is an exciting period potentially improving the therapeutic outcome of leukemia and lymphoma patients; however, there are still unanswered questions and several issues to be addressed. Clinical data on acute leukemias are still missing while results in other entities are ambiguous or obtained from rather small cohorts. Appropriate combination partners need to be identified, ideal application sequences and dosages are to be evaluated, and biomarkers are necessary to offer a useful and potent CDKI-based therapy for the special needs of every molecular subgroup in a variety of hematological entities.

## Figures and Tables

**Figure 1 cancers-13-02497-f001:**
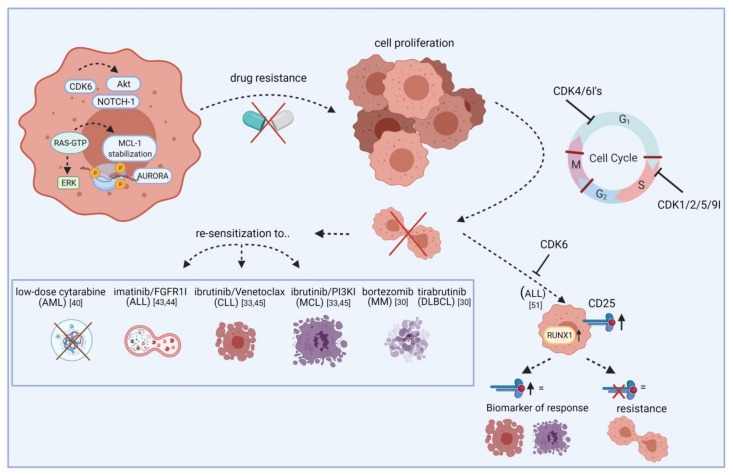
Re-sensitization of drug-resistant cells by CDK inhibitors. The scheme illustrates how CDKIs can conquer resistance of leukemic blasts towards targeted agents. Leukemic cells show frequent overexpression or constitutive activation of oncogenic kinases like AURORA and AKT. While these kinases mediate drug resistance via apoptosis inhibition or increased tumor cell proliferation, blocking specific CDKIs may help to make cells more vulnerable to certain approved drugs, including cytarabine, imatinib, and ibrutinib. This effect is likely mediated by cell cycle checkpoint blockade, resulting in impaired cell division and proliferation. Additionally, blocking CDKs induces CD25 abundance on ALL cells and this may act as a biomarker for response. Vice versa, lack of CD25 expression indicates resistance. FGFR1I, fibroblast growth factor receptor 1 inhibitor; PI3KI, phosphatidyl inositol 3 kinase inhibitor; MCL, mantle cell lymphoma; MM, multiple myeloma; DLBCL, diffuse large B-cell lymphoma. Created with BioRender.

**Figure 2 cancers-13-02497-f002:**
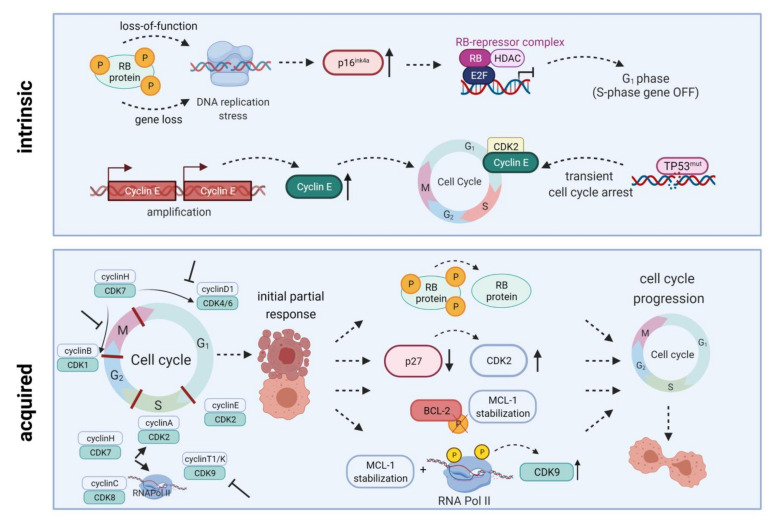
Intrinsic and acquired resistance mechanisms towards CDKIs. Intrinsic resistance is driven by lack of functional Rb protein, leading to DNA replication stress, high-level expression of endogenous CDK4/6 inhibitor p16*^INK4A^*, and inactivation of E2F-regulated genes. Another resistance mechanism is based on Cyclin E amplification and overexpression driving cell cycle progression. Quite in line, *TP53* mutations are generally associated with low response to CDK inhibition. Acquired resistance mechanisms include Rb loss, CDK2 reactivation upon p27 downregulation, a truncated BCL-2 protein lacking the phosphorylation loop domain, as well as prolonged MCL-1 stability along with RNA polymerase II phosphorylation and CDK9 kinase domain up-regulation. All these mechanisms may ultimately contribute to cell cycle progression and thus CDKI resistance. HDAC, histone deacetylase; E2F, E2F transcription factor family; RNA Pol II, RNA polymerase II. Created with BioRender.

**Figure 3 cancers-13-02497-f003:**
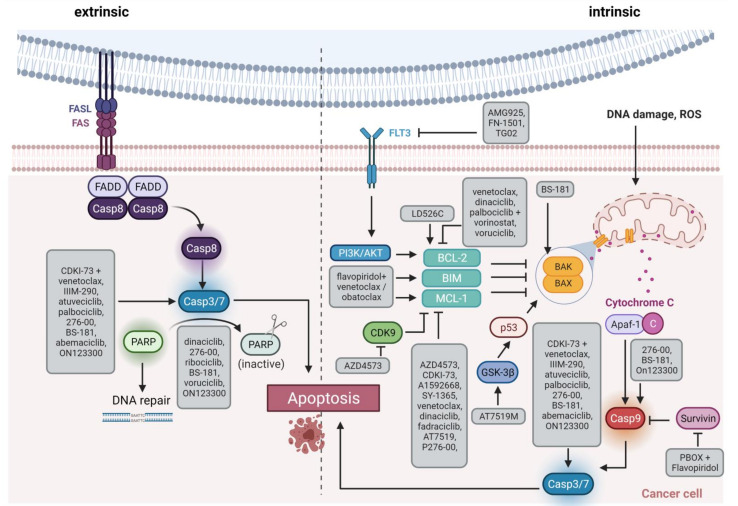
Schematic overview of apoptosis induction. The extrinsic pathway via Fas/FasL interaction is shown on the left side, and the intrinsic pathway via members of the BCL-2 family such as BAX/BAK on the right side. The highlighted boxes indicate the active form of the molecule. The gray boxes list therapeutic substances and how they influence apoptosis. FAS, Fas cell surface death receptor; FASL, FAS ligand; Casp, caspase, PARP, Poly(ADP ribose) Polymerase; FLT3, FMS related receptor tyrosine kinase 3; PI3K, phosphatidyl inositol 3 kinase; AKT, AKT serine/threonine kinase; ROS, reactive oxygen species; BCL-2, BCL2 apoptosis regulator; BIM, BCL2 like 11; MCL-1, MCL1 apoptosis regulator; BAK, BCL2 antagonist/killer 1; BAX, BCL2 associated X, apoptosis regulator; GSK-3β, glycogen synthase kinase 3-beta; p53, tumor protein 53; Apaf-1, Apoptotic peptidase activating factor 1. Created with BioRender.

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
