# Peer review of "Cyclin-Dependent Kinase Inhibitors in Hematological Malignancies—Current Understanding, (Pre-)Clinical Application and Promising Approaches"

_cancers, 2021, doi:10.3390/cancers13102497_

Round 1
Reviewer 1 Report
Please see the attached file for detailed comments.

Author Response
We thank reviewer 1 for the careful revision of our manuscript as well as the many suggestions for improvement. We have now revised the review and think that it greatly benefits from the corrections made by reviewer 1. We hope that it is now suitable for publication in Cancers.
First of all, we would like to thank reviewer 1 for the excellent suggestion in issue (3), asking to re-structure the manuscript and sort it by inhibitors instead of disease. In order to increase the clarity of the revised manuscript, this was not highlighted or marked in color because otherwise the entire text would be affected.
We are now separately addressing the issues pointed out by reviewer 1:
(1) The current structure of the review makes it difficult to follow. The different hematological malignancies are listed as separate sections, under which the effects of CDKIs are described largely in cell lines, mouse models and in clinical studies. Several inhibitors are repeated under each type of malignancy, and because their mechanisms of action are more or less the same, this results in unnecessary repetition. For example, Palbociclib is mentioned that it reduces proliferation, triggers apoptosis and reduces RB phosphorylation, G1 arrest…etc and this is repeated under AML, ALL, CML, MCL, anaplastic large cell lymphoma…etc. What is unique for every type of malignancy is the combination therapy such as Palbociclib and danusertib or MK-2206 in AML, Palbociclib and Imatinib or the FGFR1 inhibitor in ALL, …etc
Answer: We absolutely agree with the reviewer and have now re-structured the manuscript. It is now sorted by inhibitors and the molecular mechanism of each drug is now explained only once, followed by a description of the preclinical and clinical studies conducted with the agent. We believe that this change is a fundamental improvement and results in better clarity and less repetition.
(2) Similarly, the common mechanism of CDK9Is such as AZD4573 that mediates its action by inducing apoptosis, suppression of MCL-1…etc is also repeated several times under different hematologic malignancies. The same applies to the combination of AZD4573 and the BCL2-inhibitor venetoclax, which is repeated at least 3 times in AML, ALL and DLBCL.
Answer: This is also true and as already explained in issue (1), we have now fully restructured the manuscript to avoid those repetitions.
(3) To avoid repetitions, I suggest the authors structure the review around the class/category of CDKIs, rather than the hematologic malignancy. For example, a section with the title CDK4/6 inhibitors, which includes Palbociclib, ribociclib, abemaciclib and the recent lerociclib. Another section could be dedicated to Pan-CDK inhibitors, a third to CDK9Is…etc.
Answer: We once again thank for this suggestion and have performed the requested changes. The manuscript is now structured as follows: 1. Introduction; 2. CDK4/6 inhibitors; 3. CDK7/8/9 inhibitiors; 4. Pan CDK inhibitors; 5. Dual kinase inhibitors and novel approaches; 6. Conclusions.
(4) After each section/topic, it is recommended to write a couple of sentences to conclude this topic and to link it to the following section.
Answer: We have added a paragraph after each section, summarizing the respective field and highlighting open questions. Those sentences are marked in red color in the revised manuscript:
CDK4/6 inhibitors: “CDK4/6 inhibitors including the novel promising molecule lerociclib are without doubt the most intensively researched group of CDKIs in both, solid and hematological tumors. While palbociclib, ribociclib and abemaciclib already obtained FDA approval for breast cancer, current preclinical and clinical studies for leukemia and lymphoma also focus on inhibitors of other CDKs.” We believe that the final sentence is a good bridge to the following paragraph on CDK7/8/9 inhibitors which are currently heavily investigated.
CDK7/8/9 inhibitors: “CDK7/8/9 inhibitors are an emerging field and offer a very specific possibility to modify transcription processes and apoptosis regulation. Other approaches featuring down-regulation of these kinases are less specific, resulting in a broader spectrum of mechanistic actions.” We included a respective sentence to link to the following section, pan CDK inhibition, which includes inhibitors that also inhibit CDKs 7, 8 or 9 but further target a variety of other CDKs.
Pan CDK inhibitors: “Due to its variety in mechanistic pathways, pan CDK inhibition offers a wide range of possibilities to target nearly every hematological malignancy. Especially dinaciclib is already investigated in several clinical trials; however, results are ambiguous. Chemical modifications or combinatorial schedules might be beneficial to increase the therapeutic outcome.” With this concluding paragraph we aim to emphasize the great potential but also major pitfalls of pan CDK inhibition. The following section is on dual kinase inhibitors and novel approaches like PROTACS, so the last sentence should be a good introduction to this part.
Dual kinase inhibitors and novel approaches: “Novel and innovative strategies like dual kinase inhibition or PROTAC design are important steps towards an improved targeted therapy. Further preclinical studies, especially in animal and xenograft models, are needed to identify those candidates that are most likely to prove beneficial in a clinical setting.” We believe that this part demonstrates the importance and variability of innovative strategies. At the same time, it is important to mention that those approaches are not yet clinically tested and that their therapeutic potential must be evaluated carefully.
(5) As there are hundreds of papers that investigate the effect of CDKIs on cell lines, more focus on clinical studies and PDX mouse models is recommended.
Answer: There are indeed numerous preclinical studies investigating CDKIs and their effects on cell lines and basic in vivo systems. During the initial preparation of the review we have already restricted our paper to the most promising candidate substances where only preclinical data was available. We believe that the old structure of the manuscript, where the content was sorted by disease instead of inhibitor subgroups, resulted in the repeated mentioning and explanation of molecular mechanisms obtained from in vitro models. This gave the impression of an unbalanced high amount of cell line input. With the structure of the review updated, we believe that the important clinical data as well as meaningful PDX studies are more in focus.
(6) The review needs to be more balanced as it appears more focused on AML in comparison to other types of hematologic malignancies.
Answer: Reviewer 1 is correct in his/her notion that there is a higher amount of data on AML compared to other subtypes. This is simply explained by the fact that most research on CDKIs was conducted in AML models, both preclinically and clinically. In order to include relevant information in the same depth as for the other entities, more space is needed for AML compared to the other malignancies. A second explanation is once again the old (and now improved) structure of the manuscript. As AML was the first entity to be addressed, all molecular background information regarding the mechanism of action of the inhibitors was explained in detail in this section. This conveys the impression of focusing on AML. We hope that the updated structure of the manuscript resulted in a more balanced distribution.
(7) Please elaborate in the description of Figure 2 so that the legend would include a more comprehensive overview of how CDKIs re-sensitize blasts to targeted agents.
As suggested we have improved the figure legend (now Figure 1) and hope that it features now the desired degree of detail: “Figure 1. Re-sensitization of drug-resistant cells by CDK inhibitors. The scheme illustrates how CDKIs can conquer resistance of leukemic blasts towards targeted agents. Leukemic cells show frequent overexpression or constitutive activation of oncogenic kinases like AURORA and AKT. While these kinases mediate drug resistance via inhibition of apoptosis or increased tumor cell proliferation, blocking specific CDKIs may help to make cells more vulnerable to certain approved drugs, including imatinib, ibrutinib and cytarabine. This effect is likely mediated by cell cycle checkpoint blockade, resulting in impaired cell division and proliferation. Also, blocking CDKs induces CD25 abundance on ALL cells and this may act as biomarker for response. Vice versa, lack of CD25 expression indicates resistance. Created with biorender.”
(8) In lines 113-116, it is mentioned that lerociclib is superior to Palbociclib regarding efficacy and doesn’t cause neutropenia in dog models. Would you please elaborate what makes this novel inhibitor unique in not causing neutropenia?
Answer: Thanks to reviewer 1 for pointing out this missing information. This important characteristic of lerociclib should be mentioned in more detail as it offers a solution to the most important pitfall of CDK4/6 inhibitors. Hence, we included an explaining paragraph: “The novel CDK4/6 inhibitor lerociclib (G1T38) was capable of decreasing Rb phosphorylation and proliferation and induced G1 cell cycle arrest in leukemia and lymphoma cell lines in vitro. Of note, lerociclib demonstrated a superior efficacy compared to palbociclib and did not induce severe neutropenia in an estrogen receptor positive breast cancer dog animal model, which is usually a common side effect of CDK4/6 inhibitors. Using mouse xenograft models, the authors further demonstrated that lerociclib accumulated within the tumor but not in plasma, implying less severe effects on myeloid progenitor cells compared to palbociclib therapy. Indeed, plasma concentrations after palbociclib treatment were significantly higher than the concentration needed to inhibit cell proliferation in several cell lines while levels were lower after lerociclib treatment. The authors thus claim that palbociclib-induced neutropenia is a product of CDK4/6 inhibition in the bone marrow, preventing the proliferation of healthy bone marrow cells [59].”
(9) More emphasis should be placed on mechanisms of resistance to CDKIs. They were mentioned briefly, such as loss of RB function and down regulation of p27. However, since this is an important topic affecting the clinical success of CKIs, I would include it as a separate section with more details and potential strategies to overcome resistance.
Answer: We absolutely agree with reviewer 1 that the investigation of resistance mechanisms are very important for the clinical application and should be explained in this review in great detail. Unfortunately, there is only very limited data available and we have summed up every study researching these mechanisms in hematological malignancies. Using figure 2 we explain known ways of intrinsic and acquired resistance establishment. In addition, known mechanisms for all inhibitors are described directly after the presentation of preclinical studies and clinical trials. Combinatorial approaches or dual kinase inhibitors are promising strategies to overcome resistance. This is mentioned in a separate “Dual kinase inhibitors and novel approaches” section: “Due to resistance observed in CDK inhibitor mono therapy in vitro, in vivo and in clinical trials, novel approaches are being evaluated, including dual kinase inhibitors.”
(10) One aspect that is missing in the review is the interaction of CDKIs with the tumor microenvironment, especially in leukemias where leukemia stem cells reside in the hypoxic niches of the bone marrow. How does hypoxia, or other factors in the bone marrow interact with CDKIs?
Answer: We agree that this is an interesting and important point. We repeated our extensive literature search and identified two papers that examined the influence of CDKIs on cells and their respective microenvironment. The first, a 2012 Leukemia paper by Johnson et al., was already included in our manuscript: “In CLL, dinaciclib downregulates MCL‑1 gene and protein expression and potently induces apoptosis in patient-derived cells [93]. Interestingly, this effect was also present when the cells were cultivated with cytokines produced by microenvironment cells but not when cultured with direct stromal cell contact. This lack of therapeutic efficacy was overcome by addition of PI3Kα inhibitor PIK‑75 but not inhibitors of other PI3K isoforms [93].“ A second study by Cosimo et al. (Clinical Cancer Research, 2013) investigated the effect of the roscovitine derivate CR8 on CLL cells cultured alone or in stromal co-culture. The authors show that CR8 is capable of inhibiting tumor cell proliferation also in stromal co-culture but do not present any detailed mechanistic analyses. As this pan CDK inhibitor was not further analyzed in following studies and no clinical trials were conducted and due to limited space availability, we decided to not include this work in the manuscript.
However, as this topic is of great interest and relevance, we still nevertheless included a separate paragraph on microenvironment interaction, highlighting this area as an important topic that needs further investigation:
“6. The interaction of CDKIs with the tumor microenvironment
A major challenge for receiving long-term disease free-survival in hematological diseases is successful targeting of the malignant niche in the bone marrow. The latter is a complex microenvironment in which hematopoietic stem cells interact with multiple non-hematopoietic cell types. Just like normal stem cells, leukemia stem cells, hosted in the stem cell niche, undergo self-renewal, can efflux drugs, and have a quiescent cell cycle status, which makes them difficult to target. Especially CDK6 plays a critical role in hematopoietic stem cell differentiation [129]. A comparable role in leukemia stem cells can be expected.
Ischemia-like conditions are the driving force of leukemia stem cell refractoriness to classical drugs as well as targeted agents, such as imatinib mesylate [130]. Though experimental evidence for the interaction of CDKI’s with hypoxic niches of the bone marrow is still pending, experiences from solid tumor models described successful reversal of hypoxia-mediated therapy resistance via CDK1 and CDK2 interaction with hypoxia inducible factor‑1 [131]. The mechanism was due to Rb hypophosphorylation by palbociclib, which favors cell cycle arrest and senescence. A recent case report identified rebound lymphocytosis in a CLL patient after terminating palbociclib treatment for synchronic breast cancer [132]. This report impressively demonstrates the interaction of CDK4/6 inhibitors with the tumor microenvironment via induction of a cell cycle arrest. Still, follow-up studies should focus on the stem cell niche to judge the efficacy of CDKI’s and ultimately prevent relapse.”
(11) The conclusion needs to address open research questions in the field and provide potential ideas for the future research direction to bridge the current knowledge gaps in the field. Perhaps the authors could discuss the pros and cons and provide an opinion on the use of each of the categories of CDKIs in the conclusion. For example, the use of pan-CDK inhibitors versus specific CDKIs, CDK4/6Is vs CDK9Is…etc
Answer: Thanks to reviewer 1 for pointing out this missing information. Discussing the use of one class of CDKI against another is very difficult at this point because there is no clinical data at all for acute leukemias and CML. For other entities only one class of inhibitors has been tested (only pan CDK inhibition for CLL). Still, preclinical data is promising for nearly every inhibitor in each entity. We therefore decided to implement a text passage in the conclusion addressing this issue and discussing it as further research direction as suggested by the reviewer. The following passages have been inserted in the conclusion:
“The results of extensive preclinical evaluations are promising with several clinical studies currently testing CDKIs in most hematological malignancies. CDK4/6Is palbociclib, abemaciclib, and ribociclib are already FDA-approved for solid neoplasms, raising hopes for a subsequent clinical implementation in hematological malignancies in the near future. Interestingly, clinical trials have mainly been finished for lymphoma (especially MCL) and CLL patients while data on acute leukemias is lagging behind. Published results indicate that combined therapy with other small molecule inhibitors like ibrutinib or bortezomib are beneficial and can increase the therapeutic efficacy of CDKIs. Importantly, most clinical studies so far have been conducted in heavily pretreated relapsed/refractory patient cohorts, demonstrating the potential of CDK targeting for hematological malignancy management.
Another open question that must be addressed in future studies is why inhibitors of the same class result in highly variable response rates in the same entity. This underlines the importance of further investigation of biomarkers, resistance mechanisms and exact modes of action.
This is an exciting period potentially improving the therapeutic outcome of leukemia and lymphoma patients; however, there are still unanswered questions and several issues to be addressed. Clinical data on acute leukemias is still missing while results in other entities are ambiguous or obtained from rather small cohorts. Appropriate combination partners need to be identified, ideal application sequences and dosages are to be evaluated and biomarkers are necessary to offer a useful and potent CDKI-based therapy for the special needs of every molecular subgroup in a variety of hematological entities.”
Minor issues:
(1) Please check some frequent typos/grammar such as cytorabine (it should be cytarabine), malignant instead of malign (line 40), despite, not despite of (line 69), and others.
Answer: We apologize for these unnecessary errors. We have corrected the mentioned mistakes as well as other repeated errors like BCR-ABL1 instead of BCR-ABL, CDK instead of CKD or CDKIs instead of CDKI’s.
(2) Please increase the font in the bottom panel of Figure 3.
Answer: We apologize for the inconvenience and have modified the figure (now Figure 2) accordingly.
Reviewer 2 Report
The manuscript entitled “Cyclin-Dependent Kinase Inhibitors in Hematological Malignancies – Current Understanding, (Pre-)Clinical Application and Promising Approaches” is a review of the use of cyclin-dependent kinase inhibitors in the treatment of hematological malignancies. The review contains appropriate information and sites many specifics. It is reasonably well organized but is poorly written and would benefit from careful proofreading. Reviews should simplify and support concepts. The use of cyclin-dependent kinase inhibitors is an extremely complex and technical area of investigation and therefore, innately difficult to simplify. Because of this it is extremely important that the authors convey their ideas in a succinct, cohesive, and understandable manner. There is a tremendous amount of detailed information provided but it is not well molded into a contiguous flowing manuscript. It comes across as being composed of snippets from primary sources. This makes the manuscript extremely difficult to read. Additionally, the text looks like alphabet soup, increasing this difficulty. Many of the substances (enzymes) are commonly referred to by abbreviations (many of which are not defined). Because of the sheer number of pathways that are affected this makes it difficult for the reader. Unfortunately, this is confounded by inclusion of many other abbreviations, which in many cases could be avoided. Avoiding their use would increase the clarity of the text. A list of abbreviations should be included.
It would be informative for the authors to include some discussion as to why certain inhibitors of one class (ie. CDK4/6Is) are used as opposed to similar inhibitors of the same class, and what the current thinking is on why some are more effective (if they are) than others.
Addressing these issues will make this manuscript less confusing and easier to read. The authors need to address these concerns to increase the clarity and impact of this work.
.
Author Response
We thank reviewer 2 for the careful revision of our manuscript as well as the many suggestions for improvement. We have now revised the review and think that it greatly benefits from the corrections made by reviewer 2. We hope that it is now suitable for publication in Cancers.
First of all, we would like to note that the entire manuscript has been re-structured and is now sorted by inhibitor subclasses instead of entities. In order to increase the clarity of the revised manuscript, this was not highlighted or marked in color because otherwise the entire text would be affected.
We are now separately addressing the issues pointed out by reviewer 2:
The manuscript entitled “Cyclin-Dependent Kinase Inhibitors in Hematological Malignancies – Current Understanding, (Pre-)Clinical Application and Promising Approaches” is a review of the use of cyclin-dependent kinase inhibitors in the treatment of hematological malignancies. The review contains appropriate information and sites many specifics.
Answer: We thank reviewer 2 for this positive feedback.
It is reasonably well organized but is poorly written and would benefit from careful proofreading. Reviews should simplify and support concepts. The use of cyclin-dependent kinase inhibitors is an extremely complex and technical area of investigation and therefore, innately difficult to simplify. Because of this it is extremely important that the authors convey their ideas in a succinct, cohesive, and understandable manner.
Answer: We apologize for this impression by reviewer 2. There were indeed many longer sentences which complicated the understanding of the topic. We have carefully revised the manuscript and simplified those passages.
There is a tremendous amount of detailed information provided but it is not well molded into a contiguous flowing manuscript. It comes across as being composed of snippets from primary sources. This makes the manuscript extremely difficult to read.
Answer: We agree that the initial manuscript was rather difficult to read. After fully re-structuring the review, we believe that it is now better understandable. A big advantage of the new structure is the fact that general mechanisms of action of each inhibitor group are now explained at the beginning of each section. This should increase the clarity of the manuscript and improve the reading flow a lot.
Additionally, the text looks like alphabet soup, increasing this difficulty. Many of the substances (enzymes) are commonly referred to by abbreviations (many of which are not defined). Because of the sheer number of pathways that are affected this makes it difficult for the reader. Unfortunately, this is confounded by inclusion of many other abbreviations, which in many cases could be avoided. Avoiding their use would increase the clarity of the text. A list of abbreviations should be included.
Answer: We agree with reviewer 2 that the text includes a lot of abbreviations. We have now avoided many of them, for example HR+, HSC, MCL, MM, NHL, GMP, PFS, TKI, FGFR1, DLBCL, BTK, H3K27, MDS, ER, FL, SLL, MZL, HL or EZH2. Regarding the substances, there are no abbreviations that were introduced by us. During drug development, a novel candidate is usually assigned a “letter-and-number”-name which we typically included in brackets. We did this because early preclinical papers often refer to these terms. Reviewer 2 also correctly marked that some of the pathway member abbreviations are not explained. Examples are AKT, ERK or MAPK. In our opinion, these terms, as well as most protein and gene names, are well established and known by the majority of the readership. Including the long gene/protein name in the text would enlarge sentence length and make the text hard to follow. As suggested by the reviewer, we have included a list of abbreviations that also lists the full names of all genes, proteins and pathways mentioned in the manuscript.
It would be informative for the authors to include some discussion as to why certain inhibitors of one class (ie. CDK4/6Is) are used as opposed to similar inhibitors of the same class, and what the current thinking is on why some are more effective (if they are) than others.
Answer: We thank reviewer 2 for this suggestion and implemented a respective paragraph disussing the (preclinical) superiority of CKD4/6 inhibitor lerociclib over palbociclib: “The novel CDK4/6 inhibitor lerociclib (G1T38) was capable of decreasing Rb phosphorylation and proliferation and induced G1 cell cycle arrest in leukemia and lymphoma cell lines in vitro. Of note, lerociclib demonstrated a superior efficacy compared to palbociclib and did not induce severe neutropenia in an estrogen receptor positive breast cancer dog animal model, which is usually a common side effect of CDK4/6 inhibitors. Using mouse xenograft models, the authors further demonstrated that lerociclib accumulated within the tumor but not in plasma, implying less severe effects on myeloid progenitor cells compared to palbociclib therapy. Indeed, plasma concentrations after palbociclib treatment were significantly higher than the concentration needed to inhibit cell proliferation in several cell lines while levels were lower after lerociclib treatment. The authors thus claim that palbociclib-induced neutropenia is a product of CDK4/6 inhibition in the bone marrow, preventing the proliferation of healthy bone marrow cells [59].”
As clinical data availability for all inhibitors and entities is rather low, it is extremely difficult to give statements on which drug is superior over another and what the reasons for these differences are. We discussed this as open research questions in the conclusions section: “The results of extensive preclinical evaluations are promising with several clinical studies currently testing CDKIs in most hematological malignancies. CDK4/6Is palbociclib, abemaciclib, and ribociclib are already FDA-approved for solid neoplasms, raising hopes for a subsequent clinical implementation in hematological malignancies in the near future. Interestingly, clinical trials have mainly been finished for lymphoma (especially mantle cell lymphoma) and CLL patients while data on acute leukemias is lagging behind. Published results indicate that combined therapy with other small molecule inhibitors like ibrutinib or bortezomib are beneficial and can increase the therapeutic efficacy of CDKIs. Importantly, most clinical studies so far have been conducted in heavily pretreated relapsed/refractory patient cohorts, demonstrating the potential of CDK targeting for hematological malignancy management. Another open question that must be addressed in future studies is why inhibitors of the same class result in highly variable response rates in the same entity. This underlines the importance of further investigation of biomarkers, resistance mechanisms and exact modes of action.”
Addressing these issues will make this manuscript less confusing and easier to read. The authors need to address these concerns to increase the clarity and impact of this work.
Answer: We hope that we addressed all issues in sufficient depth and conciseness. If there are any more changes necessary, we are glad to further adapt the manuscript.
Reviewer 3 Report
The review: "Cyclin-Dependent Kinase Inhibitors in Hematological Malignancies – Current Understanding, (Pre-)Clinical Application and Promising Approaches" by A. Richter et al. is dealing with actual and important problem of using CDK inhibitors. It represents extensive and actual compilation of most important articles in the field and will be interesting especially for clinicians. Nevertheless, several problems prevent the acceptance of manuscript.
The quality of language which makes it difficult to read and to understand. For example:
i) repeated using of the same expressions like “grim prognosis” (lines 32 and 69)
ii) wrong construction of the sentences:
Transcription factor Foxo3a can control p27 gene expression but was not influenced in palbociclib-resistant cells (line 122); “…highly durable leukemia regression” (line 148); “It even evoked anti-tumor effects in animal models and prolonged survival” instead of “… decreased tumor volume and prolonged life span” (line 208).; In T-ALL and MLL-rearranged/BCR-ABL positive B-ALL …” (line 253); line 552: “”Hematological neoplasms comprise a wide field of entities… “
iii) wrong definitions:
“Cell lines without resistance … “instead of “sensitive cell lines”; in line 476 must be “classical Hodgkin lymphoma”, authors do not discuss other HL subtypes.
iv) wrong statements:
line 262: “Basal overexpression of CD6 and CDK4/6 inhibition resulted in G1-arrest…”
In addition,
i) each part of the review which is dedicated to different tumor entities must contain short introduction describing their biology and classification at extent sufficient to understand specific role of the inhibitors;
ii) the action of protacs must be explained using correct language as it has been done in numerous reviews.
Author Response
We thank reviewer 3 for the careful revision of our manuscript as well as the many suggestions for improvement. We have now revised the review and think that it greatly benefits from the corrections made by reviewer 3. We hope that it is now suitable for publication in Cancers.
First of all, we would like to note that the entire manuscript has been re-structured and is now sorted by inhibitor subclasses instead of entities. In order to increase the clarity of the revised manuscript, this was not highlighted or marked in color because otherwise the entire text would be affected.
We are now separately addressing the issues pointed out by reviewer 3:
The review: "Cyclin-Dependent Kinase Inhibitors in Hematological Malignancies – Current Understanding, (Pre-)Clinical Application and Promising Approaches" by A. Richter et al. is dealing with actual and important problem of using CDK inhibitors. It represents extensive and actual compilation of most important articles in the field and will be interesting especially for clinicians. Nevertheless, several problems prevent the acceptance of manuscript.
Answer: We thank reviewer 3 for this positive feedback and hope that we addressed all issues raised by reviewer 3 in sufficient depth and conciseness.
The quality of language which makes it difficult to read and to understand. For example:
i) repeated using of the same expressions like “grim prognosis” (lines 32 and 69)
Answer: We apologize for this careless mistake and have carefully revised the entire manuscript accordingly.
ii) wrong construction of the sentences: Transcription factor Foxo3a can control p27 gene expression but was not influenced in palbociclib-resistant cells (line 122);
Answer: We have modified the indicated sentences as well as other passages of the manuscript to increase clarity and make the text easier to understand: “Transcription factor FOXO3A can also control p27 gene expression. However, it was not influenced in cells demonstrating palbociclib resistance, ruling out the possibility of FOXO3A mediated p27 downregulation.”
“…highly durable leukemia regression” (line 148);
Answer: We have modified the indicated sentences as well as other passages of the manuscript to increase clarity and make the text easier to understand: “Combined therapy with AZD4573 and venetoclax induced continuous inhibition of leukemic cell proliferation in all animals, even in models intrinsically resistant to both substances (diffuse large B-cell lymphoma model SU‑DHL‑4; AML model OCI‑AML3) [62].”
“It even evoked anti-tumor effects in animal models and prolonged survival” instead of “… decreased tumor volume and prolonged life span” (line 208).;
Answer: During re-structuring of the manuscript, this sentence was deleted and the information was included in a more general statement: “By inhibiting CDKs 1, 2, 5 and 9, dinaciclib reduces cell viability, induces apoptosis and cell cycle arrest in ALL and AML cell lines, also with MLL rearrangement, primary patient cells and in vivo models [89–92].”
In T-ALL and MLL-rearranged/BCR-ABL positive B-ALL …” (line 253);
Answer: We have modified the indicated sentences as well as other passages of the manuscript to increase clarity and make the text easier to understand: “In T-ALL cells as well as B-ALL cells featuring MLL or BCR-ABL1 rearrangements, palbociclib decreases cell growth via G1 arrest and Rb dephosphorylation both in vitro and in vivo [14,25–28].”
line 552: “”Hematological neoplasms comprise a wide field of entities… “
Answer: We have modified the indicated sentences as well as other passages of the manuscript to increase clarity and make the text easier to understand: “The field of hematological neoplasms is very heterogeneous with each entity featuring individual characteristics and challenges.”
iii) wrong definitions:
“Cell lines without resistance … “instead of “sensitive cell lines”;
Answer: We have modified the indicated sentences as well as other passages of the manuscript to increase clarity and make the text easier to understand: “Cell lines sensitive to palbociclib also exhibited a drop in p27 gene expression but not protein expression.”
in line 476 must be “classical Hodgkin lymphoma”, authors do not discuss other HL subtypes.
Answer: We have inserted this information: “Of note, this is the only clinical trial so far investigating CDKIs in classical Hodgkin lymphoma.”
iv) wrong statements: line 262: “Basal overexpression of CD6 and CDK4/6 inhibition resulted in G1-arrest…”
Answer: This long sentence was indeed easy to mis-interpret because of a missing comma and its sheer length. We therefore divided the sentence and hope that it is now clear: “They found significant basal overexpression of CDK6 in B‑ALL cells. Subsequent inhibition of CDK4/6 resulted in G1 phase arrest, reduced cell proliferation and apoptosis induction accompanied by Rb dephosphorylation.”
In addition,
i) each part of the review which is dedicated to different tumor entities must contain short introduction describing their biology and classification at extent sufficient to understand specific role of the inhibitors;
Answer: We agree with reviewer 3 that the initial manuscript had major shortages in explaining the basic mechanisms of the inhibitors as well as entity biology. Due to the re-structuring of the review we think that this issue is now addressed in an appropriate way. At the beginning of the CDK4/6 inhibitor as well as the CDK7/8/9 inhibitor sections, there is now a paragraph explaining the most biological context. On the other hand, we would like to point out that the main scope of this manuscript is not the explanation of the biological background of CDKs and their regulation. The focus of the present work was to demonstrate the preclinical and clinical effects of pharmacological CDK inhibition.
ii) the action of protacs must be explained using correct language as it has been done in numerous reviews.
Answer: We regret that reviewer 3 is not in agreement with the terms and language of the description of PROTACs. We have revised the respective paragraph and now explain PROTACs as follows: “These synthetic molecules comprise a ligand for the protein of interest, for example CDK6, and another ligand which recruits E3 ligases. E3 ligases then induce ubiquitination and proteasomal degradation of the target structure.”
Round 2
Reviewer 1 Report
Dear authors,
Thank you for responding to my review comments and for restructuring the manuscript.
I suggest to please leave a space (empty line) before the summary paragraph after each section so that it stands out. Additionally, I would remove the title that is written on top of Figures 1 and 2, since it is already included in the figure legend.
Reviewer 2 Report
This revision of the manuscript is much improved. The authors have made a significant attempt to clarify the data presented and to improve the flow and readability of the manuscript. These revisions improve the manuscript tremendously. There are still a few typos, and misspelled words and therefore would benefit from careful proofreading.
Reviewer 3 Report
The authors approprietly addressed all my comments.